# Comparison of World Health Organization and Demographic and Health Surveys data to estimate sub-national deworming coverage in pre-school aged children

**Nathan C. Lo**[1]*, **Ribhav Gupta**[2], **David G. Addiss**[3], **Eran Bendavid**[4,5], **Sam Heft-Neal**[6], **Alexei Mikhailov**[7], **Antonio Montresor**[7], **Pamela Sabina Mbabazi**[7]

**1** Deparment of Medicine, University of California, San Francisco, San Francisco, California, United States of America, **2** Department of Epidemiology and Population Health, Stanford University School of Medicine, Stanford, California, United States of America, **3** Focus Area for Compassion and Ethics, Task Force for Global Health, Decatur, Georgia, United States of America, **4** Division of Primary Care and Population Health, Stanford University School of Medicine, Stanford, California, United States of America, **5** Center for Health Policy and Center for Primary Care and Outcomes Research, Stanford University School of Medicine, Stanford, California, United States of America, **6** Center on Food Security and the Environment, Stanford University, Stanford, California, United States of America, **7** Department of Control of Neglected Tropical Diseases, World Health Organization, Geneva, Switzerland

* nathan.lo@ucsf.edu

**Data Availability Statement:** DHS data files are provided publicly by DHS MEASURE (https://

## Abstract

### Background

The key metric for monitoring the progress of deworming programs in controlling soil-transmitted helminthiasis (STH) is national drug coverage reported to the World Health Organization (WHO). There is increased interest in utilizing geographically-disaggregated data to estimate sub-national deworming coverage and equity, as well as gender parity. The Demographic and Health Surveys (DHS) offer a potential source of sub-national data. This study aimed to compare deworming coverage routinely reported to WHO and estimated by DHS in pre-school aged children to inform global STH measurement and evaluation.

### Methodology

We compared sub-national deworming coverage in pre-school aged children reported to WHO and estimated by DHS aligned geospatially and temporally. We included data from Burundi (2016–2017), Myanmar (2015–2016), and the Philippines (2017) based on data availability. WHO provided data on the date and sub-national coverage per mass drug administration reported by Ministries of Health. DHS included maternally-reported deworming status within the past 6 months for each child surveyed. We estimated differences in sub-national deworming coverage using WHO and DHS data, and performed sensitivity analyses.

### Principal findings

We compared data on pre-school aged children from 13 of 18 districts in Burundi (N = 6,835 in DHS), 11 of 15 districts in Myanmar (N = 1,462 in DHS) and 16 of 17 districts in the

dhsprogram.com/data) and WHO data can be accessed publicly from the Preventive Chemotherapy and Transmission Control databank, the Preventive Chemotherapy Data Portal or the Global Health Observatory (https://who.int/neglected_diseases/preventive_chemotherapy/sth/en). All files used to conduct the analysis and corresponding instructions will be made available online (https://github.com/NathanLo3/Publication-codes).

**Funding:** NCL, RG, and DGA report funding from the World Health Organization (https://www.who.int/) for this project. World Health Organization provided previously collected data yet played no role in study design, data analysis, decision to publish, or preparation of the manuscript.

**Competing interests:** NCL, RG, and DGA report funding from the World Health Organization. AM, AM, and PM are employed by the World Health Organization.

Philippines (N = 7,594 in DHS) following data exclusion. The national deworming coverages estimated by DHS in Burundi, Myanmar, and the Philippines were 75.5% (95% CI: 73.7%-77.7%), 47.0% (95% CI: 42.7%-51.3%), and 48.0% (95% CI: 46.0%-50.0%), respectively. The national deworming coverages reported by WHO in Burundi, Myanmar, and the Philippines were 80.1%, 93.6% and 75.7%, respectively. The mean absolute differences in district-level coverage reported to WHO and estimated by DHS in Burundi, Myanmar, and the Philippines were 9.5%, 41.5%, and 24.6%, respectively. Across countries, coverage reported to WHO was frequently higher than DHS estimates (32 of 40 districts). National deworming coverage from DHS estimates were similar by gender within countries.

## Conclusions and significance

Agreement of deworming coverage reported to WHO and estimated by DHS data was heterogeneous across countries, varying from broadly compatible in Burundi to largely discrepant in Myanmar. DHS data could complement deworming data reported to WHO to improve data monitoring practices and serve as an independent sub-national source of coverage data.

### Author summary

Soil-transmitted helminths are parasitic worms that infect an estimated 1 billion people globally, primarily in low- and middle-income countries. One of the main strategies to reduce the prevalence of these parasitic infections is through preventive chemotherapy—the periodic widespread treatment of an entire at-risk population (e.g. children) with albendazole or mebendazole. The World Health Organization (WHO) tracks the progress of mass deworming programs by estimating the national coverage of these programs through aggregation of local health district program reports, which represent the proportion of at-risk people in a country that are reported to have received deworming treatment. In this study, we used an alternative source of data from the Demographic and Health Surveys (DHS) that provides maternally-reported child deworming treatment to estimate coverage in pre-school aged children and was subsequently compared to data reported to WHO. We conducted this comparison of deworming coverage at a sub-national level for three countries with available data: Burundi, Myanmar, and the Philippines. Coverage reported to WHO and estimated by DHS were broadly consistent in Burundi, highly discrepant in Myanmar, and varied in agreement for the Philippines. These differences may be explained by variability in the medication distribution monitoring practices, quality of national validation systems, and mis-reporting due to albendazole receipt through non-STH related public health campaigns (e.g. lymphatic filariasis treatment). This analysis suggests that in specific settings, DHS data can complement deworming coverage data reported to WHO, serving as an independent source of data to improve monitoring and evaluation of deworming programs globally.

## Introduction

Soil-transmitted helminthiasis (STH) affects an estimated one billion people worldwide, and is caused by infection with *Ascaris lumbricoides*, *Trichuris trichiura*, or either of the hookworm

species, *Ancylostoma duodenale* or *Necator americanus* [1,2]. Globally, the disease burden of STH disproportionately affects children in low- and middle-income countries, but STH also affects adolescents and adults [2–5]. The burden of STH is associated with a range of clinical morbidities including abdominal pain, anemia, malnutrition, and in cases of heavy STH infection, physical (e.g. growth) and possible developmental effects, some of which may be addressed through treatment with albendazole or mebendazole [1,6–9]. Preventive chemotherapy, which involves the periodic, coordinated, large-scale administration of single-dose albendazole or mebendazole (i.e., deworming treatment) to groups at risk, is the primary public health strategy to control STH burden by both treating existing infections and decreasing worm burden to prevent future morbidity [4,10]. School-aged child deworming is typically conducted through the existing school-based infrastructure [11,12]. However, since pre-school aged children cannot access these platforms, pre-school aged deworming is primarily conducted through campaign-style Child Health Days [13,14]. In 2017 alone, deworming campaigns distributed an estimated 600 million doses of medication against STH to children in endemic countries, where approximately 188 million of the doses were delivered to pre-school aged children (1–4 years) [15].

The key metric for monitoring the progress of deworming programs remains national drug coverage (%) in school-age and pre-school aged children, as reported to the World Health Organization (WHO) [3,4,16]. The ongoing public health goal set against STH by WHO is to achieve a minimum of 75% annual national coverage in both school-aged and pre-school aged children by 2020 for all countries where STH remains endemic [4,10]. Deworming coverage is reported to WHO from national Ministries of Health and published at the national-level in the Preventive Chemotherapy and Transmission Control (PCT) databank, the Preventive Chemotherapy Data Portal and the Global Health Observatory (GHO) [17–19]. There is a growing interest in utilizing sub-national data to develop programmatic metrics to better understand geographic variation, gender parity, and other equity metrics. In addition, there is an unmet need to integrate secondary datasets for data quality control into routine measurement and evaluation efforts [20–23].

The Demographic and Health Surveys (DHS) provide routinely collected survey data that may provide a complementary source of sub-national deworming data in regions endemic to STH [24]. DHS are large, nationally representative surveys of person-level data on a wide range of health measures from low- and middle-income countries designed in coordination with governmental organizations, such as a National Statistical Office or Ministry of Health [24]. As DHS are standardized across countries and are routinely conducted approximately every five years, they allow for the systematic tracking and comparison of key indicators of deworming program success over time and geography [25]. In particular, DHS collect maternally-reported data on recent deworming receipt for their pre-school aged children [24,25]. Recently, empirical studies have analyzed sub-national coverage and equity of deworming programs using DHS data [6,23,26].

Although prior analyses have compared coverage reported to WHO at the national-level with coverage estimated from specialized validation surveys by external public health institutes, no studies have evaluated the use of sub-national coverage [27,28]. The DHS offer a potential source of routinely collected, sub-national deworming data to improve coverage metrics from existing deworming program evaluation data, although questions remain as to how these DHS coverage estimates will compare to data reported to WHO. To inform future measurement and evaluation efforts for deworming programs, we performed an empirical analysis comparing sub-national deworming coverage of pre-school aged children using data routinely reported to WHO annually and as estimated by DHS.

## Methods

### Overview

We calculated deworming coverage in pre-school aged children at the primary administrative level, referred to as "district" throughout the study, using data reported to WHO and measured by DHS. As deworming coverage is campaign- and time-specific, we refer to the deworming campaign coverage for a specific year and geographic area as the "deworming coverage" throughout the study (e.g. 2016 Burundi national deworming campaign coverage is referenced as "Burundi national-level deworming coverage"). We included data from three countries (Burundi, Myanmar, and the Philippines) based on availability of temporally overlapping sub-national data reported to WHO and by DHS. The coverage data were matched geospatially (at district-level) and temporally (aligning DHS administration dates with the recall periods following campaigns reported to WHO). Recall period, the time period during which reports of treatment in DHS are likely to align with a specific deworming campaign, was defined as 6 months, since in the DHS, mothers were asked if their child received deworming during the previous 6 months. This produced a pair of coverage estimates (one DHS estimated and one reported to WHO) only for the subset of districts in each country where 1) a campaign was reported to WHO and 2) any DHS responses were collected within the recall period following the campaign reported to WHO. The coverage analysis was calculated independently for each country and compared across data sources at the district-level.

### Data source

We reviewed a complete list of countries with available district-level deworming coverage data available from WHO (using PCT and GHO databanks) and DHS (using Phase VII surveys). We included countries based on the following inclusion criteria: i) country was endemic for STH (determined by WHO recommendations for a national deworming program); ii) availability of national and district-level deworming campaign coverage data reported to WHO; and iii) availability of individual-level DHS data for the corresponding time period (defined by recall period of interviews conducted within the 6 months following campaigns reported to WHO as per DHS question format used in all countries surveyed) for at least a subset of districts reporting to WHO. Based on these criteria, we included data from each state of Burundi for 2016–2017, province of Myanmar for 2015–2016 and administrative region of the Philippines for 2017.

WHO publishes age-stratified, campaign coverage data at the administrative level one (equivalent to district-level) for preventive chemotherapy programs against STH, lymphatic filariasis, onchocerciasis, schistosomiasis, and trachoma in the Preventive Chemotherapy and Transmission Control databank, Preventive Chemotherapy Data Portal and Global Health Observatory [17–19,29]. WHO data were reported from each country's Ministry of Health, and utilize programmatic data derived from district health facilities records that are reported to their corresponding districts [17]. For each campaign, WHO data provided deworming coverage by age group (e.g. pre-school aged children), the date of campaign initiation, and the district covered. Coverage reported to WHO was computed as the total number of individuals reported to have received deworming medications over the total population of targeted pre-school aged children per district for a single deworming campaign with adjustments to align with DHS data [30]. We assumed all deworming medications reported to WHO for a deworming campaign were received within the month of deworming campaign initiation. We excluded data from any deworming campaigns with a reported coverage to WHO over 95% from the calculation of district-level coverage given concerns regarding the data reliability, although we varied this reliability threshold in the sensitivity analysis [27]. We established the

data reliability threshold value of 95% from a previous analysis of validation surveys by the U. S. Centers for Disease Control and Prevention for deworming campaigns from 2000–2011, where 0 of 16 surveys conducted had estimated coverage values above 95% despite reports to WHO for 7 of the 16 campaigns [27]. In addition, WHO provided data on the presence of and dates for campaigns initiated for lymphatic filariasis at a sub-district-level in countries endemic to lymphatic filariasis, as treatment campaigns for both STH and lymphatic filariasis utilize albendazole (applicable to children 2–5 years).

DHS are nationally representative surveys conducted across countries at the household- and individual-level amongst women and men with women 15–49 years of age as the primary respondents. We used a question asked of mothers in the DHS women's module on whether each of their children received "drugs for intestinal parasites in the last 6 months", which served as a maternally-reported metric to estimate pre-school aged child deworming coverage receipt [31]. The DHS question did not distinguish between deworming due to programmatic (i.e., deworming campaigns) and non-programmatic sources (e.g., local clinic, pharmacy, informal health practitioners) [32,33]. We excluded child deworming receipt data for children below one year of age at the estimated time of the deworming campaign (based on WHO definition of a pre-school child as 1–4 years) or when age was not specified. As surveys were conducted over an extended period in each country, we also excluded DHS observations if the date of survey administration fell outside the recall period following any campaigns reported to WHO for the district of residence, since these observations were not associated with a known source of programmatic coverage. The maternally-reported response to child deworming status was coded as "yes", "no", or "maybe"; the "maybe" status was assigned a 0.5 probability of receipt in the main analysis and varied in the sensitivity analysis.

## Statistical analysis

The primary study outcome was the reported deworming coverage by data source. We estimated pairs of deworming coverages and the mean absolute difference in district-level coverage reported to WHO and estimated by DHS for the subset of districts per country with both DHS data and data reported to WHO. We estimated the district-level coverage reported to WHO with the following procedure. First, for each deworming campaign reported to WHO, we determined if any children in the DHS dataset were eligible to have been exposed based on the date and location from where mothers were surveyed. Any campaigns with no eligible DHS observations were excluded from data reported to WHO. Second, we estimated the treatment probability for each child targeted for treatment based on coverage data reported to WHO (e.g. 75% coverage would equal 0.75 probability). For districts where only one deworming campaign reported to WHO overlapped with the recall period following DHS sampling, we used the corresponding campaign coverage as the treatment probability for each child targeted for treatment. For districts where the recall period for two deworming campaigns reported to WHO overlapped with DHS administration dates, we used a "treatment correlation" parameter, $\alpha$, to combine coverage reported to WHO from all applicable campaigns into a single treatment probability for all children targeted for treatment (See Appendix). This ensured that we generated one coverage estimate reported to WHO for each district, regardless of the number of campaigns that met inclusion criteria. This "treatment correlation" parameter, $\alpha$, represents the correlation between coverage in two sequential deworming campaigns. The "treatment correlation" parameter was calibrated to each country to minimize the difference in district-level coverage reported to WHO and estimated by DHS to provide an optimistic assumption of compatibility. Given that this treatment correlation parameter is unknown due to the absence of prior estimates in the literature, we varied this parameter in sensitivity

                                             

analyses. Third, to represent the annual district-level deworming coverage reported to WHO, we aggregated child treatment probabilities at a district-level divided by the total number of children reported to WHO to have been targeted for deworming. Finally, to calculate the national-level deworming coverage using data reported to WHO, we utilized a population-weighted average of coverage across all districts.

We estimated the district-level deworming coverage by DHS data with the following procedure. First, we assigned a treatment probability to each child in the DHS dataset based on maternal report of her child's deworming status. Second, we estimated the district-level coverage by calculating the mean probability of deworming receipt using maternally-reported child deworming status per district. We estimated the standard deviation of the district-level DHS coverage estimation by assuming a binomial distribution of deworming receipt given the binary treatment of all survey response options for deworming receipt status. Finally, we estimated the national-level deworming coverage and standard deviation by calculating the mean probability of deworming receipt using DHS data across all districts. For national-level coverage estimates, we applied DHS-provided survey weights, due to the multi-stage cluster sampling procedure, which are intended to estimate statistics representative at the national-level from sample data [24]. We estimated gender-stratified, national-level and district-level coverages using the same methodologies outlined.

This study relied upon data de-identified during collection by DHS, and data provided at the district-level from WHO. The study met all requirements for review exemption by the University of California, San Francisco Institutional Review Board. All analyses were conducted using the R programming language version 3.8.0 (R Foundation for Statistical Computing, Vienna, Austria) and the Stata/IC Software version 15.1 (StataCorp LP, College Station, United States). The analysis files are available online [34].

## Sensitivity analyses

We conducted a series of sensitivity analyses on key parameters and assumptions to determine the robustness of our findings and test potential explanations for differences across countries. We adjusted DHS survey inclusion by varying the recall period assumed for maternally-reported response (base case: 6 months) across a short-term period of 4 to 8 months that we believed would still yield reliable DHS responses to child deworming receipt to evaluate scenarios when the true maternal recall might have been shorter or longer than designated by the actual survey question. We also varied the campaign reliability threshold (base case: exclude WHO data with >95% coverage) to test exclusion of campaigns with >90% coverage to no exclusion based on reported coverage. We varied the "treatment correlation" parameter to test scenarios with a 70% correlation between the populations targeted by overlapping deworming campaigns reported to WHO. We performed an analysis of coverage differences across data sources, after exclusion of children surveyed by DHS below 2 years of age, stratified by district-level treatment of lymphatic filariasis using albendazole and Diethylcarbamazine (applicable to children 2–5 years) to serve as a proxy to assess the effect of misclassification due to non-deworming public health programs also distributing albendazole (see Appendix). Finally, we varied the probability of deworming receipt for "maybe" responses in the DHS (base case: +0.5) to test +0 to +1 deworming receipt probability and assess the effect on DHS coverage estimates.

## Results

### Descriptive results

We included DHS observations collected in Burundi from October 2016 to March 2017, Myanmar from December 2015 to July 2016, and the Philippines from August 2017 to October

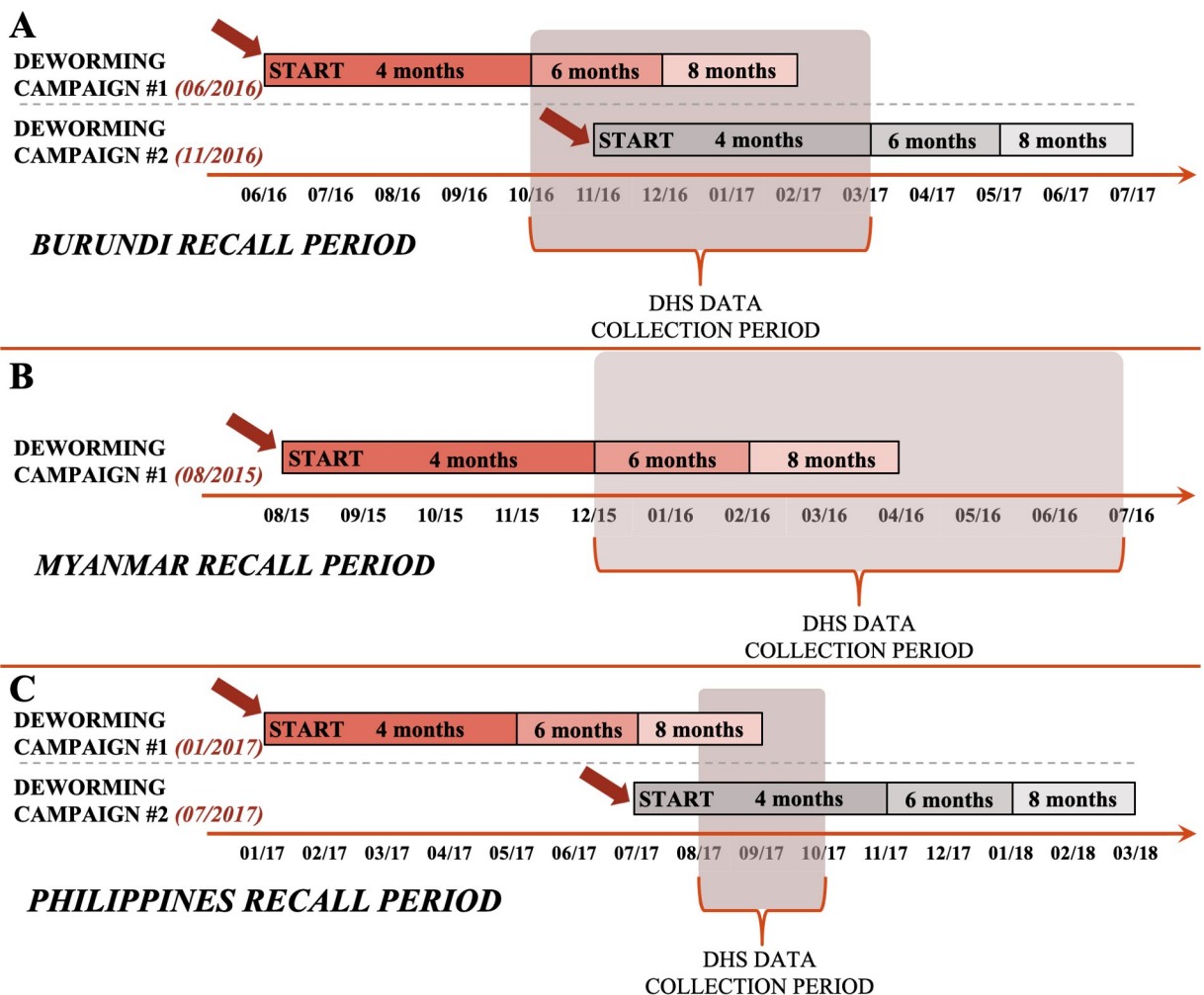

**Fig 1. Timeline of national deworming campaigns reported to WHO for pre-school aged children and DHS data collection periods in study countries.** We obtained sub-national deworming coverage data reported to WHO and estimated by DHS across comparable timelines. We used WHO data on programmatic deworming to determine national deworming campaign start dates (dates on left-hand side and time-axis in month/year, e.g. 06/2016 is June 2016) and estimated coverage for Burundi (panel A), Myanmar (panel B) and the Philippines (panel C). We used DHS data from each country to estimate deworming coverage by matching respondents to the corresponding deworming campaigns, based on the survey date and recall period in the orange/gray bars (base, 6-month recall; alternatively, 4- or 8-months based on the sensitivity analysis) that represents the accepted time period (months) after deworming campaign start.

2017 (see Fig 1). DHS observation periods corresponded to deworming campaigns reported to WHO for Burundi during June 2016 and November 2016, for Myanmar during August 2015, and for the Philippines during January 2017 and July 2017 (Fig 1).

We estimated deworming coverage using data reported to WHO and estimated by DHS for 13 of 18 districts of Burundi, 11 of 15 districts of Myanmar, and 16 of 17 districts of the Philippines. The data from remaining districts were excluded from analysis based on criteria for data unreliability defined as coverage reported to WHO > 95% (N = 5 districts in Burundi, N = 3 districts in Myanmar, N = 1 in the Philippines) or absence of any DHS data for a given district (N = 1 district in Myanmar). After data exclusion, the final study sample size for DHS maternally-reported data was 6,835 pre-school aged children in Burundi (50.2% female), 1,462 pre-school aged children in Myanmar (48.3% female), and 7,594 pre-school aged children in Philippines (47.6% female; See Figs 2 and 3 and Appendix).

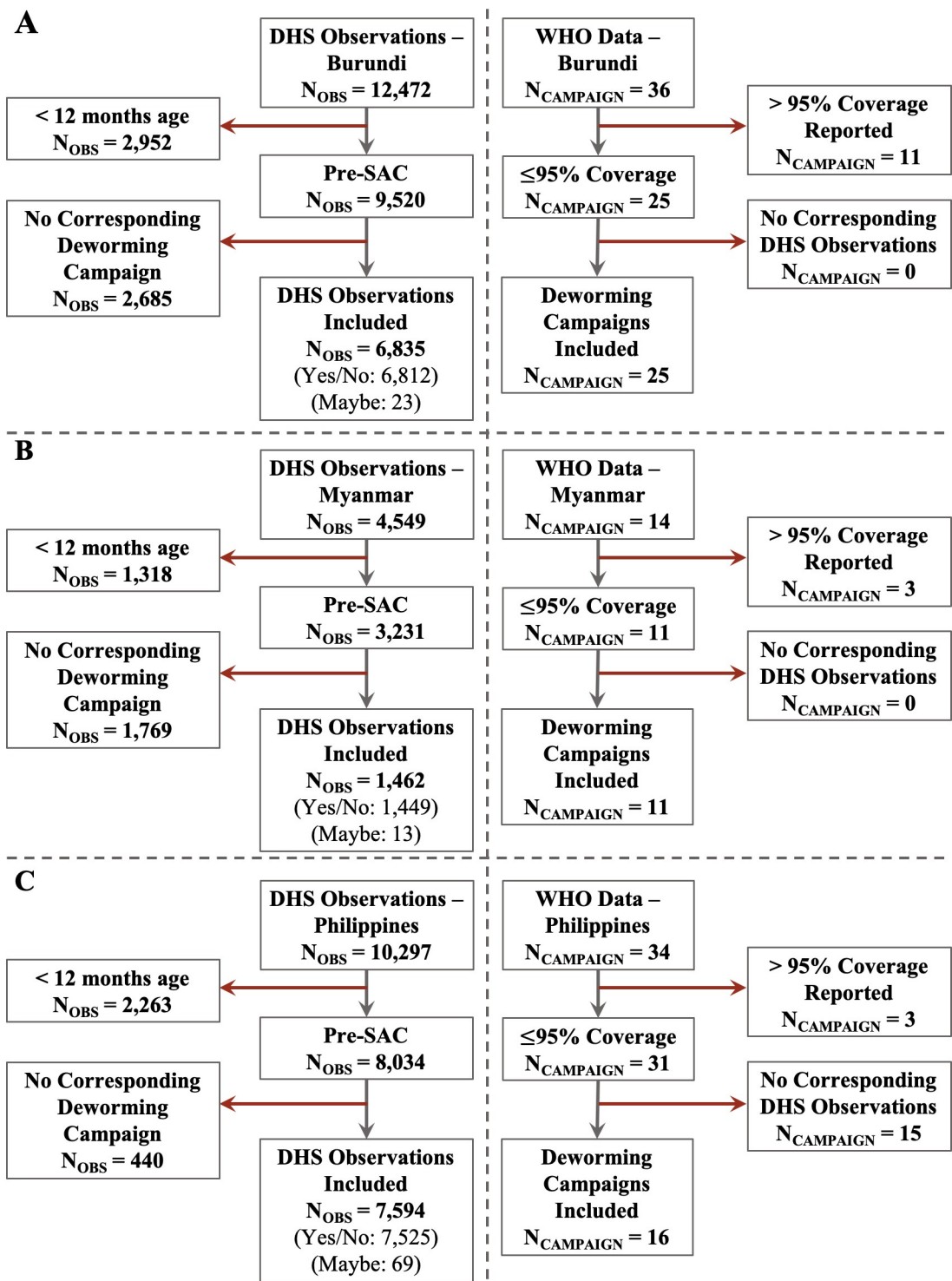

**Fig 2. Inclusion and exclusion criteria for DHS and WHO deworming data of pre-school aged children for study countries.** The data inclusions and exclusion are provided for the three study countries: (A)Burundi; (B) Myanmar; and (C) Philippines. Key definitions: $N_{OBS}$: Number of pre-school aged children sampled in DHS; $N_{CAMPAIGN}$: Number of deworming campaigns reported to WHO; Pre-SAC: Pre-school aged children (12–59 months). In DHS observations, the response of "yes/no" refers to children who were or were not dewormed, while "maybe" refers to mothers who were unable to recall.

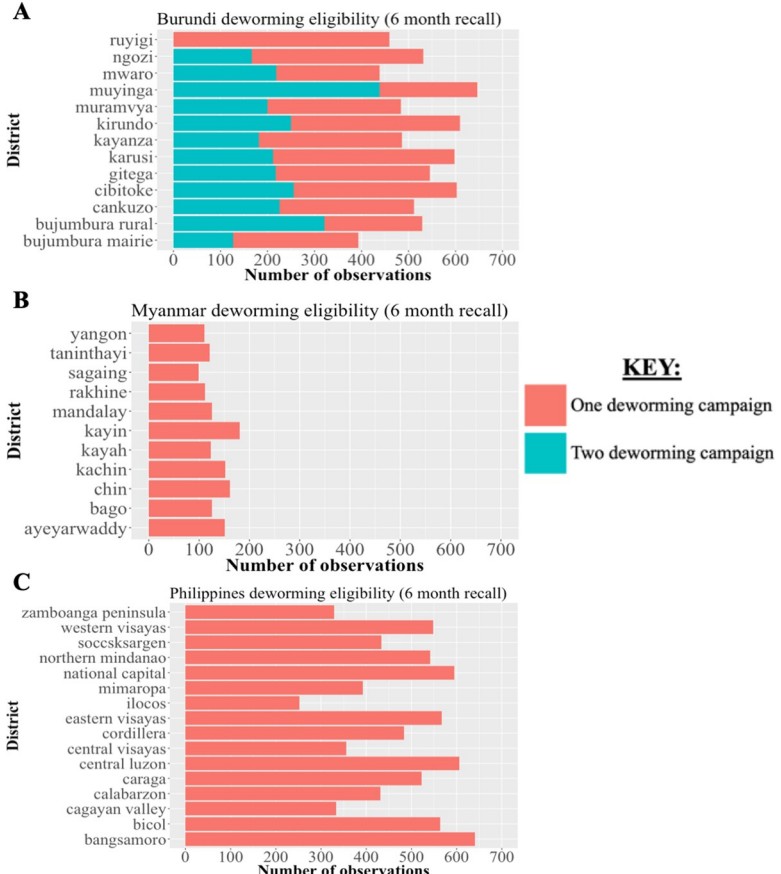

**Fig 3. District-level sample size of DHS observations used to estimate deworming coverage in study countries.** We determined the sample size of DHS respondents at a district-level and estimated the proportion eligible for one or two deworming campaigns based on the maternal recall period used for DHS observation inclusion. These data are visualized for Burundi (panel A), Myanmar (panel B), and the Philippines (panel C) each with a base case recall period of 6 months based on formulation of the original DHS question for deworming receipt.

## Main analysis

In Burundi, using data reported to WHO, the estimated national-level deworming coverage was 80.1% with district-level estimates ranging from 55.1% (Bujumbura Mairie) to 91.1% (Bujumbura Rural) (See Fig 4). Using DHS deworming data, the estimated national-level deworming coverage was 75.5 (95% CI: 73.3%-77.7%) with district-level estimates ranging from 58.1% (Cibitoke) to 88.6% (Kayanza). Stratifying coverage by child gender, using DHS deworming data the estimated national-level deworming coverage for girls was 74.6% and was 76.5% for boys of pre-school age (See Fig 5). The mean absolute difference in district-level deworming coverage reported to WHO and estimated by DHS data was 9.5% with absolute differences ranging from 0.9% (Kirundo) to 32.4% (Cibitoke). Coverage estimates reported to WHO were higher in 9 of 13 districts included.

In Myanmar, using data reported to WHO, the estimated national-level deworming coverage was 93.6% with district-level estimates ranging from 88.4% (Kayah) to 94.9% (Mandalay). Using DHS deworming data, the estimated national-level deworming coverage was 47.0% (95% CI: 42.7%-51.3%) with district-level estimates ranging from 29.8% (Sagaing) to 71.1% (Kayah). Stratifying coverage by child gender, using DHS deworming data the estimated national-level deworming coverage was 49.2% for girls and 44.8% for boys of pre-school age.

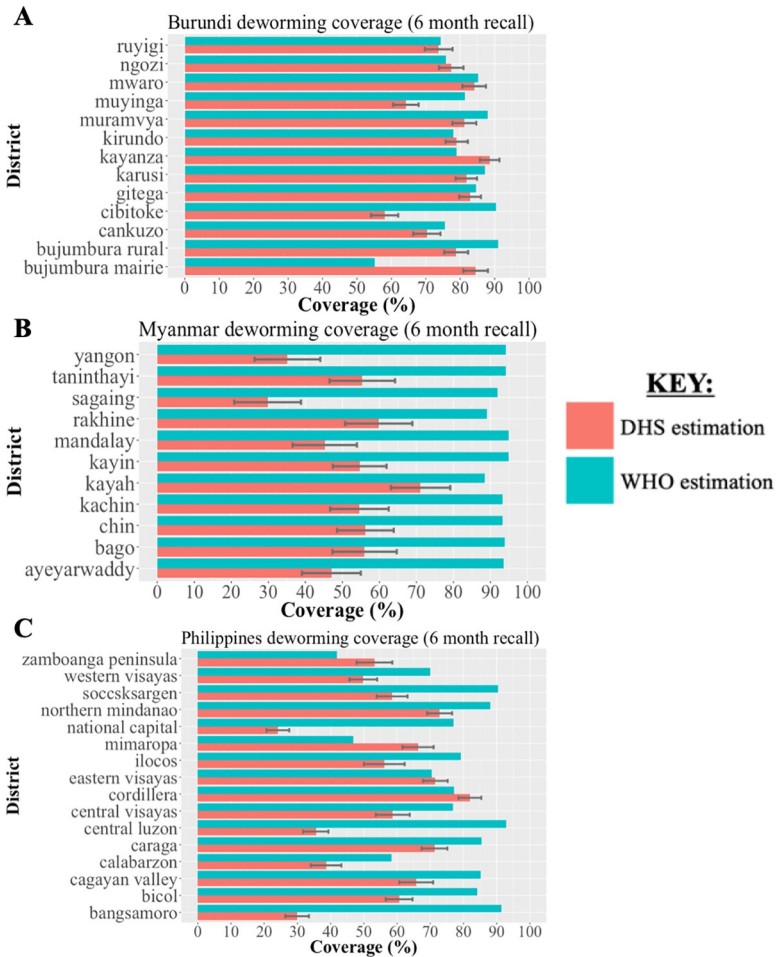

**Fig 4. Comparison of district-level deworming coverage in pre-school aged children reported to WHO and estimated by DHS data for study countries.** We obtained WHO data on district-level deworming coverage reported by national Ministries of Health based on health facilities data. We estimated district-level deworming coverage from DHS data corresponding to the known WHO deworming campaigns using maternally-reported data on child deworming receipt. The comparison of deworming coverage using WHO and DHS data are shown for a base case recall period of 6-month in Burundi (panel A), Myanmar (panel B) and the Philippines (panel C) based on the DHS question formulation.

The mean absolute difference in district-level deworming coverage reported to WHO and estimated by DHS data was 41.5% with absolute differences ranging from 17.3% (Kayah) to 62.2% (Sagaing). Coverage estimates reported to WHO were higher in all 11 districts included.

In the Philippines, using data reported to WHO, the estimated national-level deworming coverage was 75.7% with district-level estimates ranging from 42.0% (Zamboanga Peninsula) to 92.9% (Central Luzon). Using DHS deworming data, the estimated national-level deworming coverage was 48.0% (95% CI: 46.0%-50.0%) with district-level estimates ranging from 24.1% (National Capital) to 82.9% (Cordillera). Stratifying coverage by child gender, using DHS deworming data the estimated national-level deworming coverage for girls was 48.4% and was 47.6% for boys of pre-school age. The mean absolute difference in district-level deworming coverage reported to WHO and estimated by DHS data was 24.6% with absolute differences ranging from 1.1% (Eastern Visayas) to 61.5% (Bangsamoro). Coverage estimates reported to WHO were higher in 12 of 16 districts included.

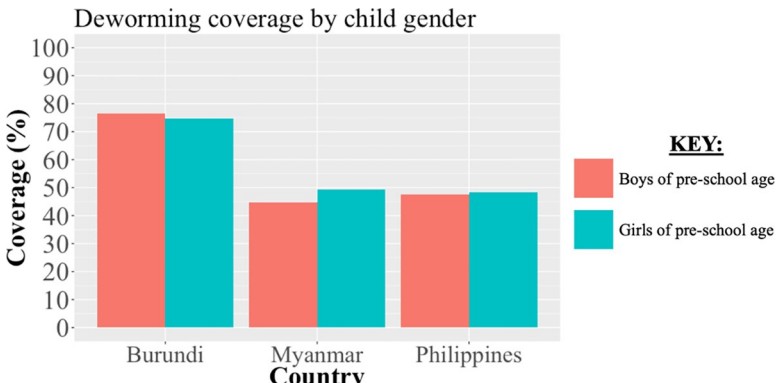

**Fig 5. Comparison of national-level deworming coverage amongst girls and boys of pre-school age estimated by DHS data in study countries.** We estimated the national-level deworming coverage using maternally-reported child deworming status from DHS stratified by child gender (girls and boys). We compared the estimated coverage by gender using a base recall period of 6 months in Burundi, Myanmar, and the Philippines as based on the DHS question formulation.

National-level deworming coverage reported to WHO was higher in all three countries studied—Burundi, Myanmar, and the Philippines—compared to coverages estimated using DHS data. Cumulatively across all three countries, district-level coverage reported to WHO was higher than coverage estimated using DHS data for 32 of the 40 districts included.

## Sensitivity analyses

Varying the assumed maternal recall period for deworming receipt in DHS data, which served as inclusion criteria for DHS observations, had minimal impact on the sample size and the estimated coverage for Burundi and the Philippines, but had larger effects on the sample size and coverage estimates using DHS data in Myanmar (see Appendix). Varying the data reliability threshold, used to exclude WHO reported campaigns, had minimal effects on the mean absolute differences in district-level deworming coverage in Burundi and the Philippines but had large effects in Myanmar. Varying the "treatment correlation" parameter, used to estimate treatment probability for children exposed to multiple deworming campaigns reported to WHO, had large effects on the national-level coverage using data reported to WHO for Burundi but had no effect on coverage estimates for Myanmar and the Philippines, as in the latter countries all DHS observations aligned with a single campaign given a 6-month recall period. In Burundi, varying the "treatment correlation" parameter from 70% (optimized scenario of data compatibility) to 100% correlation between treatment by overlapping deworming campaigns reported to WHO changed the mean absolute difference in district-level deworming coverage from 10.1% to 9.6%.

## Discussion

In this study, we used datasets from WHO and DHS to compare sub-national deworming coverage estimates for pre-school aged children in Burundi, Myanmar and the Philippines. We found the results to be heterogeneous; deworming coverage estimates by data source were broadly compatible for Burundi, largely discrepant for Myanmar, and varied in agreement for the Philippines. Sub-national differences in coverage across data sources presented no apparent geospatial clustering within each country. In addition, we found deworming coverage by gender to be broadly similar at the national-level within all three countries when using DHS data.

This study included data from three countries with diverse settings to highlight scenarios with different geographies, levels of STH prevalence, baseline deworming coverage, and lymphatic filariasis endemicities [35,36]. The study also highlighted key challenges to integrating DHS and WHO data for deworming coverage estimates including reliability of systems reporting to WHO, accuracy of DHS self-reported deworming receipt, and concurrent medication by other public health programs using albendazole, such as for lymphatic filariasis (where mothers may misclassify treatment as only preventive for lymphatic filariasis and not as a deworming treatment). This analysis highlighted opportunities to identify sub-national regions with low coverage that may merit further investigation. Variations in sub-national coverage per country may be attributed to the use of decentralized health systems where sub-national regions are left to apply their own methods for implementing deworming activities. Overall, the findings support the utilization of sub-national DHS deworming data under specific settings to complement data reported to WHO by serving as an independent source of geographically and demographically disaggregated coverage data for pre-school age children.

The difference in agreement of coverage estimated using data reported to WHO and from DHS for Burundi, Myanmar and the Philippines may be partially explained by variation in the quality of national data systems reporting to WHO and recall inaccuracies in the DHS survey due to self-reporting or due to misclassification of albendazole receipt from other public health programs (e.g. albendazole for lymphatic filariasis) or from non-programmatic distribution (e.g. albendazole given in clinic). One explanation may be differences in quality of data reporting and health information management protocols, which can be driven by a host of factors (e.g., financial, robustness of reporting systems) across multiple levels of data processing (e.g. health facility, districts, country level) affecting data reported to WHO. All three countries reported greater than 95% and sometimes above 100% deworming coverage to WHO for some districts, which based on prior literature are unlikely to be accurate estimates [27,28]. As district-level deworming coverage estimates reported to WHO were typically higher than estimates by DHS (32 out of 40 districts, particularly for all 11 districts of Myanmar included), this could indicate a systemic cause of under-estimation in DHS datasets (e.g., maternal under-reporting) or of over-estimation in WHO datasets (e.g., albendazole medication delivered but not administered to a child). In the case of Myanmar, regional endemicity for lymphatic filariasis potentially increased misclassification of coverage estimates by DHS data as maternally-reported deworming receipt were much lower than reports to WHO, possibly due to only associating the treatment with its indication of lymphatic filariasis (applicable for children 2–5 years) and not for deworming. While DHS data may be biased from including unprogrammed sources of deworming receipt, not present in data reported to WHO, this would bias estimates by DHS to over-estimate coverage, which would not explain the observed pattern of discrepancies. On the other hand, in Burundi, exclusion of 30% of campaigns reported to WHO due to data reliability concerns (coverage >95%) may indicate an under-reporting of the target population size (e.g. use of outdated census data) or over-reporting of doses delivered (e.g. doses delivered outside target age range) to WHO.

Our study findings present some differences from prior analyses that compared coverage reported to WHO with estimates by independent surveys. Deworming coverage validation surveys have previously been conducted by the U.S. Centers for Disease Control and Prevention and Hellen Keller International immediately following national deworming campaigns [27,28]. Both series of validation surveys used multi-stage sampling methodologies for household data collection similar to methods employed by DHS and defined coverage through similar methods to those used for estimating coverage by DHS. Both validation survey analyses found that deworming coverage estimates reported to WHO were widely discrepant from their independent estimation efforts at the national-level. Yet, comparatively, our analysis

found coverage reported to WHO and estimated by DHS data may be compatible in some settings. Although validation surveys are specifically designed to monitor deworming program success, the DHS offer a routine and standardized alternative that will likely prove cost-efficient given that deworming data are already collected as part of a larger effort (and there would not be additional cost to access this data), rather than through a specialized survey requiring additional resource allocation. Additionally, DHS would provide a dataset to track supplementary equity metrics (e.g. coverage by gender, urban/rural living, and socioeconomic status) currently unstudied given the format of data reported by national governments to WHO, and could be correlated with conventional metrics such as the Human Development Index. In settings with limited albendazole distribution outside of deworming campaigns (e.g. in districts non-endemic for lymphatic filariasis) and where districts endemic to STH reported a deworming coverage of below 90% to WHO, the likelihood for comparability with sub-national DHS data appears much higher. The broad compatibility of coverage estimated by DHS data with reports to WHO in Burundi was particularly notable given political unrest in the country at the time of the data reporting in 2016–2017 that may have disrupted public health efforts [37,38]. Gender-specific estimates of deworming coverage from DHS showed no marked differences (see Appendix), and as deworming coverage data reported to WHO becomes gender-stratified, further analyses can be undertaken to measure the agreement of gender equity metrics.

The deworming coverage data reported to WHO and estimated by DHS measure deworming using different mechanisms. Most notably, DHS deworming data are reported based on maternal recall and may include coverage by unprogrammed sources (e.g. local pharmacy, clinic) and albendazole receipt for non-STH related public health campaigns (e.g. as part of lymphatic filariasis treatment) in addition to coverage by programmed sources. As DHS data relied on maternally-reported child deworming receipt, they are potentially subject to misclassification of treatment arising from recall bias. However, recent work suggests maternally-reported child deworming data are a reliable estimate of programmatic coverage in the short-term with reported estimates within 5–10% of official campaign estimates, and with vaccine coverage estimates by WHO routinely incorporating maternally-reported survey data [39,40]. Additionally, as DHS coverage is estimated using survey sample data, non-response biases may affect the sample representativeness, which is not the case with data reported to WHO. However, as the response rates for all three countries were over 90% (with no indications of differential non-response) and survey weights were applied for national-level comparisons, we believe DHS data are likely representative at least at the national-level [41–43]. In aggregating data at multiple points from the individual health facility records up to the national-level, it is possible that records reported to WHO were re-counted or contained errors, which may both have over- or under-estimated coverage reported; however, this is not possible for DHS datasets which provided household level survey data. Similar differences in data collection methodologies have been noted in vaccine literature when comparing annual administrative and survey data on vaccine coverage. As both sources have their merits and limitations, agencies often rely on field experts to determine the most probable vaccine coverage with WHO recommendations for the periodic implementation of data quality and coverage evaluation surveys [40,44,45].

The study results should be taken in context with the limitations of the data and assumptions of the analysis (see Appendix). As only three countries were included, due to current challenges aligning DHS and WHO datasets available, the analysis presents unique scenarios of compatibility but is unlikely to be globally representative of the compatibility of data reported to WHO and by DHS. The proposed secondary source of deworming data, DHS, relied on maternally-reported child deworming receipt, which are potentially subject to recall

bias as discussed above. In selecting data, we excluded deworming campaigns reporting >95% coverage to WHO based on literature suggesting such coverage values are often inaccurate and over-estimated [27,28] although the optimal threshold for data reliability remains unclear given the high variability in the number of districts in our analysis upon varying the threshold chosen. Additionally, WHO recommends semiannual treatment in countries with higher STH endemicity, assuming a 6-month separation between campaigns. However in some cases (i.e. select districts of Burundi) campaigns were conducted within a shorter time period of one another. This may be due to alignment with other public health campaigns or anticipated seasonal variation in campaign resources. To adjust treatment probabilities for repeat coverage over multiple campaigns, a "treatment correlation" parameter $\alpha$ was derived through a fitting process that was designed to minimize differences between district-level coverage reported to WHO and estimated by DHS (i.e. the optimistic scenario). The fit parameter's value of $\alpha =$ 0.822 applied to coverage estimation for data reported to WHO from Burundi indicates we assumed 67.8% of all children covered were treated by both deworming campaigns—an assumption that is unlikely to have occurred as the parameter was fit for the most optimistic case of coverage estimate compatibility. Furthermore, sensitivity analysis found that this poorly characterized parameter will likely affect the study findings in settings with semiannual treatment and further empirical data are needed to understand these model inputs. As lymphatic filariasis coverage data are often reported for all age groups and are not specific to preschool children, misclassification of albendazole receipt for only lymphatic filariasis may have a smaller effect than found given we assumed all districts receiving treatment will include preschool aged children in coverage. Myanmar presents a case study for compatibility limitations due to a much smaller DHS sample size (<150 samples per district) than those used for Burundi or the Philippines which may limit the sub-national representation of deworming estimates from DHS data and have led to the wide discrepancies in coverage estimates by data source observed. This was due to a smaller DHS campaign, fewer children per mother interviewed, and a very limited alignment of the DHS sampling period with the recall period for deworming campaigns reported to WHO. Finally, as DHS samples are collected approximately every five years, this precludes their annual use as per conventional WHO reporting and with many DHS datasets only made available a year or longer following surveying, this limits the ability to provide real-time feedback for national deworming.

In this study, we found that district-level deworming coverage estimates for pre-school aged children reported to WHO and estimated by DHS were broadly consistent across Burundi, highly discrepant in Myanmar, and varied in alignment across the Philippines. As deworming coverage remains a key metric in monitoring and evaluating programs against STH globally, our study indicates that incorporation of DHS for deworming coverage for quality control, under specific settings, may improve data quality and resource allocation by systematically classifying the urgency of coverage validation surveys for each national reporting system. Additionally, where coverage estimates are compatible, DHS deworming data can support the development of novel deworming coverage equity metrics, such as by geography and gender.

## Supporting information

**S1 Checklist. STROBE Checklist.**
(PDF)

**S1 Technical Appendix. Additional methodologic details for estimating deworming coverage from data reported to the World Health Organization (WHO) and by using the**

**Demographic and Health Surveys (DHS) data.**
(DOCX)

**S1 Table. District-level deworming coverage in pre-school aged children using data reported to WHO and estimated by DHS in Burundi under base case analysis.**
(DOCX)

**S2 Table. District-level deworming coverage in pre-school aged children using data reported to WHO and estimated by DHS in Myanmar under base case analysis.**
(DOCX)

**S3 Table. District-level deworming coverage in children 2–4 years of age using data reported to WHO and estimated by DHS in the Philippines under base case analysis.**
(DOCX)

**S4 Table. District-level deworming coverage in pre-school aged children using data reported to WHO and estimated by DHS in Myanmar and the Philippines stratified by level of endemicity of lymphatic filariasis.**
(DOCX)

**S1 Fig. District-level sample size of DHS observations used to estimate deworming coverage in study countries, varying maternal recall period.** We estimated the sample size of DHS respondents at the district-level and estimated the proportion eligible for one or two deworming campaigns based on a maternal recall period of 6 months. This data is visualized for Burundi (panel A-B), Myanmar (panel C-D), and the Philippines (panel E-F), each with a varying maternal recall period from 4 or 8 months.
(DOCX)

**S2 Fig. Correlation between district-level deworming coverage in pre-school aged children reported to WHO and estimated using DHS data for study countries.** We obtained WHO data on district-level deworming coverage reported by national Ministries of Health based on health facility records. We estimated district-level deworming coverage with DHS data corresponding to the known WHO deworming campaigns using maternally-reported data on child deworming receipt. The comparison of deworming coverage using WHO and DHS data are shown for the base case recall period of 6-month in Burundi (panel A), Myanmar (panel B) and the Philippines (panel C) based on the DHS question formulation.
(DOCX)

**S3 Fig. Comparison of district-level deworming coverage in pre-school aged children reported to WHO and estimated by DHS in study countries, varying maternal recall periods.** We obtained WHO data on district-level deworming coverage reported by national Ministries of Health based on health facility records. We estimated district-level deworming coverage with DHS data corresponding to the known WHO deworming campaigns using maternal-reported data on child deworming receipt. The comparison of deworming coverage using WHO and DHS data is shown with varying maternal recall period from 4 months to 8 months in Burundi (panel A-B), Myanmar (panel C-D), and the Philippines (panel E-F).
(DOCX)

**S4 Fig. Comparison of district-level deworming coverage in pre-school aged children reported to WHO and estimated by DHS in study countries, varying threshold for data reliability.** We obtained WHO data on district-level deworming coverage reported by national Ministries of Health based on health facility records. We estimated district-level deworming coverage with DHS data corresponding to the known WHO deworming campaigns using

maternal-reported data on child deworming receipt. The comparison of deworming coverage using WHO and DHS data is shown with a varying data reliability threshold (base case of 95%) from > 90% to no threshold to change the districts meeting inclusion criteria in Burundi (panel A-B), Myanmar (panel C-D), and the Philippines (panel E-F).
(DOCX)

**S5 Fig. Comparison of district-level deworming coverage in pre-school aged children reported to WHO and estimated by DHS in study countries, varying the treatment correlation parameter.** We obtained WHO data on district-level deworming coverage reported by national Ministries of Health based on health facility records. We estimated district-level deworming coverage with DHS data corresponding to the known WHO deworming campaigns using maternal-reported data on child deworming receipt. The comparison of deworming coverage using WHO and DHS data for Burundi is shown with a varying treatment correlation (base case of 67.8%) value from 70% (panel A) to 100% (panel B).
(DOCX)

**S6 Fig. Comparison of district-level deworming coverage in pre-school aged children reported to WHO and estimated by DHS in study countries, varying treatment probability for "maybe" responses in DHS data.** We obtained WHO data on district-level deworming coverage reported by national Ministries of Health based on health facility records. We estimated district-level deworming coverage with DHS data corresponding to the known WHO deworming campaigns using maternal-reported data on child deworming receipt. The comparison of deworming coverage using WHO and DHS data is shown with varying receipt status for all "maybe" DHS responses (base case of +0.5) from +0 and +1 in Burundi (panel A-B), Myanmar (panel C-D), and the Philippines (panel E-F).
(DOCX)

**S7 Fig. Comparison of mean district-level deworming coverage in children 2–4 years of age reported to WHO and estimated by DHS for countries endemic to lymphatic filariasis stratified by level of endemicity.** Endemicity to lymphatic filariasis was determined by sub-district-level data reported to WHO on the record of a treatment campaign during the year of deworming analysis within each country. In Myanmar, districts were categorized as endemic to lymphatic filariasis if half of sub-districts received treatment (panel A) and in the Philippines, districts were categorized as endemic to lymphatic filariasis if any of their sub-districts received treatment (panel B). Note difference in endemicity definition used for Myanmar and the Philippines.
(DOCX)

## Author Contributions

**Conceptualization:** Nathan C. Lo, Antonio Montresor, Pamela Sabina Mbabazi.

**Formal analysis:** Nathan C. Lo, Ribhav Gupta.

**Investigation:** Nathan C. Lo, Ribhav Gupta, David G. Addiss, Eran Bendavid, Sam Heft-Neal, Alexei Mikhailov, Antonio Montresor, Pamela Sabina Mbabazi.

**Methodology:** Nathan C. Lo, Ribhav Gupta, David G. Addiss, Eran Bendavid, Sam Heft-Neal, Alexei Mikhailov, Antonio Montresor, Pamela Sabina Mbabazi.

**Writing – original draft:** Nathan C. Lo, Ribhav Gupta.

**Writing – review & editing:** Nathan C. Lo, Ribhav Gupta, David G. Addiss, Eran Bendavid, Sam Heft-Neal, Alexei Mikhailov, Antonio Montresor, Pamela Sabina Mbabazi.

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
