## [Decision Letter · Decision Letter 0]

12 Mar 2020

Dear Dr. Lo,

Thank you very much for submitting your manuscript "Comparison of World Health Organization and Demographic and Health Surveys data to estimate sub-national deworming coverage in pre-school aged children" for consideration at PLOS Neglected Tropical Diseases. As with all papers reviewed by the journal, your manuscript was reviewed by members of the editorial board and by several independent reviewers. In light of the reviews (below this email), we would like to invite the resubmission of a significantly-revised version that takes into account the reviewers' comments. 

We cannot make any decision about publication until we have seen the revised manuscript and your response to the reviewers' comments. Your revised manuscript is also likely to be sent to reviewers for further evaluation.

Sincerely,

David Joseph Diemert, M.D.

Associate Editor

Jennifer Keiser

Deputy Editor

Reviewer's Responses to Questions

**Key Review Criteria Required for Acceptance?**

**Methods**

-Are the objectives of the study clearly articulated with a clear testable hypothesis stated?

-Is the study design appropriate to address the stated objectives?

-Is the population clearly described and appropriate for the hypothesis being tested?

-Is the sample size sufficient to ensure adequate power to address the hypothesis being tested?

-Were correct statistical analysis used to support conclusions?

-Are there concerns about ethical or regulatory requirements being met?

Reviewer #1: o Please clarify in the manuscript the inclusion criteria for DHS-campaign dataset ‘pairs’.

Were campaigns reported to WHO included only if a DHS took place in the same district within 6 months after the campaign? Were DHSs included only if a campaign had taken place in the same district within the previous 6 months? Could a given district be included more than once if it had two different DHSs, each within six months of a different deworming campaign? 

What does a ‘base’ of 6 months that varied from 4-8 months mean? – what criteria were actually applied to yield this range?

Since DHSs are not done instantly, did the time windows for inclusion correspond to the beginning or the end of the DHS data collection period in the district? What if a DHS was started in the district less than 6 months after a campaign, but finished more than 6 months?

Related, what does ‘assumed recall’ of 6 months in the DHS mean? If it refers to the time period specifically listed in the DHS question asked of the respondent, there’s nothing ‘assumed’ about it. I’d recommend getting rid of or defining the term. If the length of time respondents were asked about varied by country, I would state that clearly when you quote the DHS question, and since there were just three countries, perhaps state which it was for each. Later the term ‘recall period’ recurs – similarly please define this if you need to keep it. It’s not clear to me that there’s a role here for worrying about the length of a respondent’s recall at all since the time window is clearly defined.

o The sentence “we excluded any deworming campaigns with a reported coverage to WHO over 95%...” is not clear – does this mean you excluded the involved districts from the analysis completely, or that you used the DHS data for those districts but not the WHO data somehow? Please specify in the manuscript. 

o Please state the source for the population estimates (including age-specific estimates) used as the denominators for the WHO coverage calculations, as well as the sources of the population estimates used in the DHS study designs.

o Please clarify what level of “location” campaigns were conducted on – were they district-level, multidistrict, did they sometimes include only part of a district, etc? the reader needs to have confidence that the areas covered are identical when comparing coverage values across the two data sets.

o DHSs are not conducted primarily in women. There is a men’s questionnaire, a women’s questionnaire, and a household questionnaire. It appears the deworming question is asked in the women’s questionnaire, which I think may be what you meant. Please correct.

o Data analysis: you discuss using weighting to calculate estimated national-level coverage, but there is no discussion of district-level weighting to account for DHS survey design and actual nonresponse. Please specify whether weighting was used for district-level coverage estimates and whether it was your own or DHS weights. It seems unlikely the DHS participants in each district were perfectly representative of its general population. 

o The term ‘deworming coverage’, as used throughout, is problematic without further qualification. It implies, like ‘vaccine coverage’, that at a given moment in time each individual in the country is either covered or not covered, when in truth people fall along a distribution of how long ago they were last dewormed. Deworming coverage is campaign-specific; it doesn’t make sense to talk about e.g. Burundi’s current national deworming coverage, but about its 2017 deworming campaign’s coverage – and even that only if the entire country did a campaign. More specific consistent terminology is needed. Please consider replacing throughout with something like ‘X year national deworming campaign coverage’, “Y year District Z deworming campaign coverage”. 

o I’m not clear on why an area would possibly have more than one official deworming campaign within 6 months of the same DHS. The highest WHO-recommended deworming frequency is biannual (https://www.who.int/elena/titles/deworming/en/), in areas with baseline prevalence over 50%. If you truly found many such situations, this requires some further explanation

Reviewer #2: Methodology was found in order.

**Results**

-Does the analysis presented match the analysis plan?

-Are the results clearly and completely presented?

-Are the figures (Tables, Images) of sufficient quality for clarity?

Reviewer #1: o School-age deworming programs are commonly done in schools. Please consider discussing the implications of this in light of gender-specific school attendance statistics. If these countries have significant gender gaps in school attendance but no gaps in deworming coverage, what is your interpretation of this? https://data.unicef.org/topic/education/primary-education/

o It’s not clear which dataset you’re attributing ‘recall bias’ from albendazole distribution by non-deworming programs to (although I think it’s misclassification bias, not recall bias, no?) Please specify- it seems more intuitive that maternally reported DHS data would be subject to this, not official district deworming records, so I’d expect that to have led to DHS data overestimating coverage, not the other way around. Unless you’re saying that you think mothers are systematically misclassifying deworming campaigns as LF campaigns? Would state this the first time you discuss this source of bias (in Methods) if so.

o Please fix ‘Center for Disease Control’ – it is ‘U.S. Centers for Disease Control and Prevention’.

o The discussion of maternal recall and recall bias seems problematic – the gestalt one gets is that mothers are being blamed for giving the ‘wrong’ answer, but their answers appear in the examples given to either not be wrong at all (correct knowledge that their own child got unprogrammed/unintended deworming) or be misclassification bias rather than recall bias. I suspect an individual mother’s recall of her child’s deworming status is likely to be much more accurate than official aggregate figures. Certainly in the vaccination world, surveys of individual immunization documentation yield far lower coverages than values reported to WHO (“administrative coverage”) and I think are considered more accurate.

o From what I can tell, MDAs for LF do not routinely include all pre-school-aged children (https://www.cdc.gov/mmwr/preview/mmwrhtml/mm6223a3.htm ; http://www.tropicalparasitology.org/article.asp?issn=2229-5070;year=2013;volume=3;issue=1;spage=67;epage=71;aulast=Ghosh – children under 2 years excluded). Please consider what effect this has on your calculations and interpretations.

o Not clear what your statement about ‘compatibility limitations’ in Myanmar means – are you just saying the DHS coverage estimates there had wide confidence intervals due to the small sample size and the WHO estimates fell within those, so you don’t know if the discrepancies are real? Would clarify.

o Please expand the final conclusion if possible. The reader has learned that in some countries, DHS and WHO coverage values were quite close, and in others they weren’t. Authors state that that means DHS data should perhaps be used going forward, but how exactly? Is your idea that if discrepancies are over a certain threshold, a more thorough quality review of the national reporting system is needed?

Reviewer #2: Results are clearly presented.

**Conclusions**

-Are the conclusions supported by the data presented?

-Are the limitations of analysis clearly described?

-Do the authors discuss how these data can be helpful to advance our understanding of the topic under study?

-Is public health relevance addressed?

Reviewer #1: o School-age deworming programs are typically done in schools. Please consider discussing the implications of this in light of gender-specific school attendance statistics. If these countries have significant gender gaps in school attendance but no gaps in deworming coverage, what is your interpretation of this? https://data.unicef.org/topic/education/primary-education/

o It’s not clear which dataset you’re attributing ‘recall bias’ from albendazole distribution by non-deworming programs to (although I think it’s misclassification bias, not recall bias, no?) Please specify- it seems more intuitive that maternally reported DHS data would be subject to this, not official district deworming records, so I’d expect that to have led to DHS data overestimating coverage, not the other way around. Unless you’re saying that you think mothers are systematically misclassifying deworming campaigns as LF campaigns? Would state this the first time you discuss this source of bias (in Methods) if so.

o Please fix ‘Center for Disease Control’ – it is ‘U.S. Centers for Disease Control and Prevention’.

o The discussion of maternal recall and recall bias seems problematic – the gestalt one gets is that mothers are being blamed for giving the ‘wrong’ answer, but their answers appear in the examples given to either not be wrong at all (correct knowledge that their own child got unprogrammed/unintended deworming) or be misclassification bias rather than recall bias. I suspect an individual mother’s recall of her child’s deworming status is likely to be much more accurate than official aggregate figures. Certainly in the vaccination world, surveys of individual immunization documentation yield far lower coverages than values reported to WHO (“administrative coverage”) and I think are considered more accurate.

o From what I can tell, MDAs for LF do not routinely include all pre-school-aged children (https://www.cdc.gov/mmwr/preview/mmwrhtml/mm6223a3.htm ; http://www.tropicalparasitology.org/article.asp?issn=2229-5070;year=2013;volume=3;issue=1;spage=67;epage=71;aulast=Ghosh – children under 2 years excluded). Please consider what effect this has on your calculations and interpretations.

o Not clear what your statement about ‘compatibility limitations’ in Myanmar means – are you just saying the DHS coverage estimates there had wide confidence intervals due to the small sample size and the WHO estimates fell within those, so you don’t know if the discrepancies are real? Would clarify.

o Please expand the final conclusion if possible. The reader has learned that in some countries, DHS and WHO coverage values were quite close, and in others they weren’t. Authors state that that means DHS data should perhaps be used going forward, but how exactly? Is your idea that if discrepancies are over a certain threshold, a more thorough quality review of the national reporting system is needed?

Reviewer #2: Conclusions are appropriate, and limitations are highlighted. May I suggest a mention of variability in reporting resulting from a decentralized health system, where districts are largely left on their own to implement mass deworming strategy. Variable MDA coverage may well be used as part of advocacy in enhancing service delivery, outputs and outcomes of peripheral health units.

**Editorial and Data Presentation Modifications?**

Reviewer #1: (No Response)

Reviewer #2: Minor revision in the discussion is recommended to include the comment made above.

**Summary and General Comments**

Reviewer #1: General: this is a useful, relevant paper for advancing determination of deworming coverage.

Abstract 

- Would state explicitly what time frame national deworming coverage reported to WHO is intended to represent (ideally would be the past 6 months, for comparability to the maternal report question, but regardless readers will need some sense of how these two sources would be compared)

Introduction

- Optionally, authors may also wish to consider mentioning the neurodevelopmental effects of heavy STH infestation 

- The basic issue authors are raising in the NTD world, of conflict between official coverage figures based on dose numbers administered (“administrative coverage”) and population-based survey results, has a long history in the world of immunization as well. Authors may find it useful to explore this and optionally mention parallels, since there is vast experience in that field with the issue. One link is https://www.who.int/immunization/monitoring_surveillance/routine/coverage/en/ . Also, the discussion of the advantages and disadvantages of each data source found here https://www.who.int/bulletin/volumes/87/7/08-053819/en/ may be useful.

- Use of terms

o Deworming is not a strategy for ‘eliminating’ the disease burden of STH – ‘elimination’ has specific public health definitions which I don’t believe is the goal of these programs. Perhaps ‘controlling’ / ‘minimizing’?

o Mass chemotherapy is not preventive – it’s intended to treat existing infestations in individuals

o WHO does not have an ongoing goal to achieve national coverage at any threshold in ‘all countries’ - please check WHO deworming recommendations. Probably ‘all endemic countries’ or ‘all highly endemic countries’ etc – maybe all countries above the baseline 20% prevalence threshold in the link that follows. Also, it may be important to clarify here that WHO recommendations for frequency of deworming (annual vs biannual) depend on the baseline prevalence (https://www.who.int/elena/titles/full_recommendations/deworming/en/

- Introduction should please make clear what this analysis provides that has not been done before. Have other studies sought to compare coverage from these two data sources, or other similar data sources, in other countries? Is this the first to do it on a subnational level? If, as implied in your discussion, this is the first attempt to triangulate using data already routinely collected for a different purpose rather than a purpose-built deworming coverage survey and you think its primary value is in the potential resources saved, would say that.

Reviewer #2: This paper is a highly relevant one and calls attention to the limitation of reported MDA coverage to WHO as basis for indicating progress of morbidity control in STH infections. The methods are found to be generally rigorous, and the results provide an excellent view of DHS data to complement the use of reported MDA data to WHO to help track progress.

PLOS authors have the option to publish the peer review history of their article (what does this mean?). If published, this will include your full peer review and any attached files.

Reviewer #1: No

Reviewer #2: No
---

## [Decision Letter · Decision Letter 1]

12 May 2020

Dear Dr. Lo,

Thank you very much for submitting your manuscript "Comparison of World Health Organization and Demographic and Health Surveys data to estimate sub-national deworming coverage in pre-school aged children" for consideration at PLOS Neglected Tropical Diseases. As with all papers reviewed by the journal, your manuscript was reviewed by members of the editorial board and by several independent reviewers. The reviewers appreciated the attention to an important topic. Based on the reviews, we are likely to accept this manuscript for publication, providing that you modify the manuscript according to the review recommendations. 

Sincerely,

David Joseph Diemert, M.D.

Associate Editor

Jennifer Keiser

Deputy Editor

Reviewer's Responses to Questions

**Key Review Criteria Required for Acceptance?**

**Methods**

-Are the objectives of the study clearly articulated with a clear testable hypothesis stated?

-Is the study design appropriate to address the stated objectives?

-Is the population clearly described and appropriate for the hypothesis being tested?

-Is the sample size sufficient to ensure adequate power to address the hypothesis being tested?

-Were correct statistical analysis used to support conclusions?

-Are there concerns about ethical or regulatory requirements being met?

Reviewer #1: Methods, Overview: Despite the definition now provided for “recall period”, it remains confusing throughout, because the definition doesn’t distinguish between “how far back we asked the respondent to think back in answering the question” and “how far back the respondent actually thought in answering the question”, which is the critical issue. To make things harder, I believe the actual most common understanding of that phrase is neither, but is more like “how far back the respondent can reliably remember things”. Please revise the definition to clarify that (if I’m correct) you mean the first option. Perhaps consider whether you need to create distinct terms for this and the other plausible meaning(s) you use later in the ‘sensitivity analysis’ section. E.g. “requested recall period” vs “actual recall period” etc.

Methods, “Data source” – is the assumption that all deworming meds were received within the month of campaign initiation justified? I don’t know how long these campaigns typically last, but if you can provide any reference or data to support the expectation that they typically last less than one month, would briefly to so here so it doesn’t seem so arbitrary. 

Data source: While inclusion criteria are clearer, we’re not all the way there yet. The “Data Source” paragraph does not answer: 1) Were countries included if they had matching DHS and WHO data for only part of the country? 2) Could a given district be included more than once if it had matched pairs of DHS and WHO data for two different years? (e.g. for one DHS and also its next DHS) . 

Optionally, you could also consider additionally stating (since it appears this is what happened with the Philippines) something like: “Data on different subnational regions from a single national DHS could potentially be matched with WHO data from one national campaign in one region but the next national campaign (six months later) in another region, if the DHS data collection took several months to progress from the first region to the second.”

Also optionally, for maximal clarity, you might also consider reorganizing this paragraph to state first the country-level inclusion criteria already provided, then the district-level inclusion criteria, then the individual-level criteria (currently provided later in the Methods section). 

Data source: Optionally, it would be of interest to tell us how many countries have deworming coverage data available from WHO, and how many from DHS. This would better contextualize the ultimate inclusion of only three.

Reviewer #2: Objectives and methods are clearly described. Statistical methods and analysis are well noted.

**Results**

-Does the analysis presented match the analysis plan?

-Are the results clearly and completely presented?

-Are the figures (Tables, Images) of sufficient quality for clarity?

Reviewer #1: (No Response)

Reviewer #2: Results are clearly laid out and of sufficient clarity.

**Conclusions**

-Are the conclusions supported by the data presented?

-Are the limitations of analysis clearly described?

-Do the authors discuss how these data can be helpful to advance our understanding of the topic under study?

-Is public health relevance addressed?

Reviewer #1: Discussion: “the difference in data compatibility” – confusing; I think you mean “different in coverage estimate values” or something. 

Discussion: please briefly discuss the size and direction of bias potentially introduced by the individual-level DHS observation inclusion criteria – what were the response rates for these three DHSs? Did the DHS report authors record any concerns about their representativeness? Are households who were found by and agreed to participate in DHS likely/known to be different in key demographic ways from those not found/refusing? Would those e.g. have made them more likely to have also been “found” by deworming campaigns, maybe biasing both sets of estimates upward, or not?

Discussion: In the limitations, please briefly address the inclusion of only three countries. This seems inevitable, given you need a close time match and DHS happen only every 5 years, and you were somewhat lucky to get countries in three different regions, but are these three likely to be representative of the many other countries where deworming happens? Optionally, does this difficulty finding time matches suggest that perhaps STH campaigns could be deliberately timed to closely precede an upcoming DHSs in countries where MoHs want to use DHS data for coverage validation?

Biannual – I think you mean semiannual (twice per year, not once every two years).

Reviewer #2: Conclusions are well supported by data presented, and public health relevance of the study is included.

**Editorial and Data Presentation Modifications?**

Reviewer #1: N/A

Reviewer #2: (No Response)

**Summary and General Comments**

Reviewer #1: The authors have addressed most of my comments. I have a few additional comments, mostly based on the new information provided, some relating to previous issues that are not completely addressed yet.

General: Please read over for grammar, syntax and precise meaning. E.g. constructions like in the abstract, “the progress of deworming programs to control” (should be “in controlling”). Or in the abstract conclusion section, “deworming coverage…was heterogeneous across countries”. I think you mean something more like “Matching between deworming coverages reported to WHO…was heterogeneous…” Or in the introduction, “There is growing interest to utilize” – should be “in utilizing”. Or in the Discussion: “our analysis found coverage differences…may be compatible”. (The differences aren’t compatible; perhaps the estimates are compatible or comparable.) Etc.

Introduction (and wherever else this term appears): “prior” infections – perhaps “existing” instead? Deworming is not treating infections that happened in the past and have resolved, but rather current ongoing infections.

Introduction: “the key metric…remains national treatment coverage” – perhaps ‘national campaign coverage”, since the prior paragraph was all about the campaigns and you’re referring to them here? Otherwise could be misunderstood as newly introducing the idea of clinical treatment based on symptoms.

Reviewer #2: Generally an acceptable paper for dissemination.

PLOS authors have the option to publish the peer review history of their article (what does this mean?). If published, this will include your full peer review and any attached files.

Reviewer #1: No

Reviewer #2: No
---

## [Editor Report · Decision Letter 2]

1 Jul 2020

Dear Dr. Lo,

We are pleased to inform you that your manuscript 'Comparison of World Health Organization and Demographic and Health Surveys data to estimate sub-national deworming coverage in pre-school aged children' has been provisionally accepted for publication in PLOS Neglected Tropical Diseases.

Best regards,

David Joseph Diemert, M.D.

Associate Editor

Jennifer Keiser

Deputy Editor

---

## [Editor Report · Acceptance letter]

10 Aug 2020

Dear Dr. Lo,

We are delighted to inform you that your manuscript, "Comparison of World Health Organization and Demographic and Health Surveys data to estimate sub-national deworming coverage in pre-school aged children," has been formally accepted for publication in PLOS Neglected Tropical Diseases.

Best regards,

Shaden Kamhawi

co-Editor-in-Chief

Paul Brindley

co-Editor-in-Chief
